# Humanoid-LLA: Open-Vocabulary Humanoid Whole-Body Control with Large Language Action Model

## Abstract

Enabling humanoid robots to follow open-vocabulary language instructions is critical for seamless human-robot interaction, collaborative task execution, and general-purpose embodied intelligence. While recent advances have improved low-level humanoid locomotion and robot manipulation, language-conditioned whole-body control remains a significant challenge. Existing methods often fail on compositional instructions and sacrifice either motion diversity or physical plausibility. To address this, we introduce **Humanoid-LLA**, a Large Language Action Model that maps natural language commands to physically executable whole-body motions for humanoid robots. Our approach integrates three core components: a unified motion vocabulary that aligns human and humanoid motion primitives into a shared discrete space; a vocabulary-directed controller distilled from a privileged policy to ensure physical feasibility; and a physics-informed fine-tuning stage using reinforcement learning with dynamics-aware rewards to enhance robustness and stability. Extensive evaluations in simulation and on a real humanoid platform show that Humanoid-LLA delivers strong open-vocabulary generalization while maintaining high physical fidelity, outperforming existing language-conditioned controllers in motion naturalness, stability, and execution success.

## 1 Introduction

Recent breakthroughs in Large Language Models (LLMs) (Shao et al., 2024; Bai et al., 2025) have significantly advanced capabilities in perception, reasoning, and decision making across a wide range of domains, from code generation to embodied action prediction, such as vision-language-action (VLA) (Kim et al.; Bjorck et al., 2025; Liu et al., 2025; Xu et al., 2024) models for navigation and robotic manipulation. Their success stems from scalable pretraining and discrete representations that enable complex behaviors to be composed in a data efficient manner. However, while most successes in embodied VLA have been achieved in robotic manipulation tasks, particularly those using gripper based systems, transferring these advantages to *humanoid whole body control* remains challenging due to the high degree of freedom and complex dynamics inherent in humanoid robots. Moreover, unlike robot manipulation tasks that can leverage large-scale teleoperated data, it is difficult and costly to collect substantial amounts of *physically executable* humanoid motion data. Naively training on kinematic human motion captures or limited robot datasets often results in a trade-off between language faithfulness and physical feasibility, especially under real-world perturbations.

Existing methods mainly rely on motion mimicking framework: learning text–to–human motion mappings from large human motion–text datasets and then project to robots. While convenient, retargeting optimizes in the human motion space, introducing systematic projection and kinematic mismatch errors that sacrifice precision in robot execution (He et al., 2025; Yue et al., 2025). Two-stage systems add physics-based tracking controllers (e.g., PHC) for post-hoc correction, improving feasibility but not fully recovering fine-grained, language-conditioned accuracy from end to end (Luo et al., 2023). End-to-end routes convert human dataset to humanoid datasets, yet offline policies often miss real-world stochasticity and perturbations, yielding brittle, imprecise behaviors on hardware (Mao et al., 2024; Shi et al., 2025). Distillation frameworks transfer a privileged tracking teacher to a

text-conditioned student, achieving strong physical fidelity in simulation but compressing semantics and control into a single VAE—often weakening language grounding and blurring action selection (Shao et al., 2025). Across paradigms, a persistent bottleneck remains: the scarcity of high-quality, diverse, physically grounded humanoid real-robot data, limiting precise language–robot alignment and motivating robot-centric representations under minimal real-robot supervision.

To addresses the data scarcity challenge, we reformulate humanoid whole-body control as an action generation problem within a unified human-humanoid motion vocabulary space. The core idea is to leverage the abundance of text-human motion corpora while maintaining a direct mapping to torque-level execution on the physical robot. Specifically, we begin by constructing a unified vocabulary through joint quantization of paired human motions and their retargeted humanoid counterparts, ensuring that the same discrete token corresponds to the same motion primitive across both embodiments. This results in a compact and reusable motion language that (i) benefits from the scalability of human motion datasets, (ii) remains compatible with humanoid actuation constraints, and (iii) provides a discrete interface suitable for large language model based reasoning and generation.

Based on this vocabulary, we bridge the semantic and physical gap through a process of **vocabulary directed action distillation**. First, we train a privileged teacher tracking policy in simulation to accurately follow dense, retargeted humanoid reference motions with high physical fidelity. This policy is then distilled into a student controller conditioned on discrete *motion tokens* instead of continuous trajectory references. By shifting the control paradigm from dense trajectories to a compact token sequence, this approach enables the robot to execute actions selected within the vocabulary space while maintaining dynamic robustness, contact stability, and smooth whole body coordination.

Built upon the aforementioned components, we finally train a **Large Language Action Model (LLA)** that maps open vocabulary instructions to the unified motion token sequences. The training proceeds in two stages. First, we conduct supervised fine-tuning (SFT) on large-scale text human motion datasets using our unified tokenizer. Optionally, we incorporate a motion chain of thought prompting strategy to encourage the model to perform structured reasoning before generating motion tokens. Subsequently, we apply reinforcement learning fine-tuning (RLFT) with feedback from the humanoid simulation environment. Here, a group relative policy optimization objective rewards the model for both semantic alignment with the instruction and the physical executability of the generated token sequences when rolled out by the vocabulary directed controller. This closed loop training paradigm injects crucial physical priors into the token generation process, ensuring high linguistic expressivity and motion diversity while maintaining physical feasibility.

Our framework, **Humanoid-LLA**, therefore integrates language understanding, human motion, and humanoid robot execution into a cohesive pipeline comprising three key components: (1) a unified motion vocabulary that semantically aligns motion primitives across human and humanoid embodiments; (2) a vocabulary directed action distillation process that bridges discrete tokens to physically executable control policies; and (3) a Large Language Action Model (LLA) trained via supervised fine-tuning on human motion datasets and further refined through reinforcement learning with physical feedback from the humanoid platform. Extensive evaluations in both simulation and real-world environments demonstrate compelling open vocabulary generalization capabilities while maintaining high physical fidelity.

We summarize our main contributions as follows:

- We present Humanoid-LLA, an end-to-end Large Language–Action Model that enables the first open-vocabulary text-to-humanoid whole-body control, mapping expressive natural language directly to executable humanoid actions.

- We introduce a unified motion vocabulary that aligns human and humanoid in latent space, thus enabling vocabulary-directed humanoid policy distillation and supervised finetuning Humanoid-LLA with large-scale text-human datasets.

- We further fine-tune Humanoid-LLA by augmenting text–human datasets with large-scale human motion chain-of-thought and integrating humanoid fidelity feedback from physical simulation, thereby improving language generalization and execution feasibility.

- Extensive evaluations in physical environments demonstrate that our method outperforms prior works on both physical feasibility and motion quality, culminating in successful deployment on real humanoid hardware.

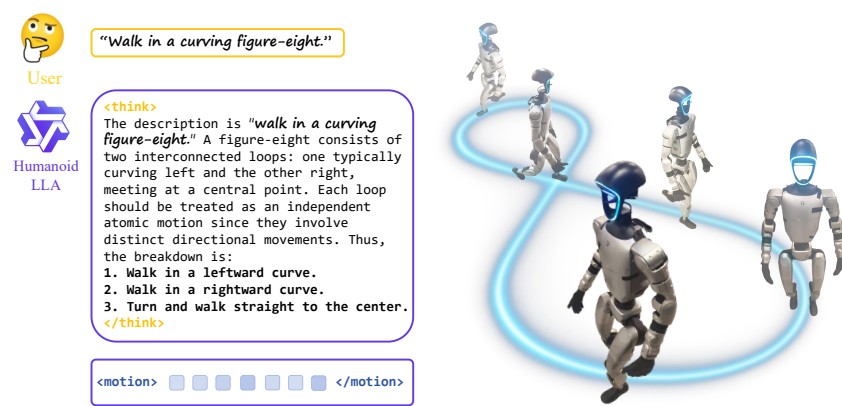

Figure 1: An illustration of Humanoid-LLA. Given an open-vocabulary instruction (e.g., "walk in a curving figure-eight"), Humanoid-LLA first use natural language (<think>) to decompose the task, then generate a sequence of unified motion tokens (<motion>). A vocabulary-directed controller executes these tokens on the robot, bridging language, a unified human–humanoid motion vocabulary, and action-level control to yield physically faithful, natural whole-body behaviors.

## 2 RELATED WORK

**Kinematic Motion Generation.** Kinematic motion generation is typically cast as conditional sequence modeling, aiming to synthesize temporally coherent pose trajectories from text, trajectories, or other control signals.Diffusion methods generate diverse, high-quality motions but are costly and hard to control(Tevet et al., 2023; Chen et al., 2023; Zhang et al., 2024; Karunratanakul et al., 2023), while GPT-based approaches improve efficiency and long-horizon consistency but are limited by quantization and data quality(Zhang et al., 2023; Jiang et al., 2024; Ouyang et al., 2025).

Recent works (Yuan et al., 2023; Serifi et al., 2024; Han et al., 2025) introduces physics priors: PhysDiff (Yuan et al., 2023) projects diffusion outputs into physically valid states via simulation, while RobotMDM (Serifi et al., 2024) integrates physical feasibility into training through reward surrogates and RL controllers. These efforts highlight the trade-off between visual fidelity and physical realism. Motivated yet distinct, we employ hierarchical physical rewards to finetune a latent motion generator via RL, and ultimately leverage a tracking policy conditioned on these latents to roll out highly physically feasible humanoid motions in simulation.

**Physics-based Character Animation.** Physics-based controllers have advanced realistic character animation, with DeepMimic (Peng et al., 2018) pioneering RL-based motion imitation and later works like AMP (Peng et al., 2021) and ASE (Peng et al., 2022) enhancing robustness and compositionality. Recent approaches adopt a generative view, such as MaskedMimic for motion inpainting and MaskedManipulator for goal-conditioned loco-manipulation (Tessler et al., 2024; 2025).

Language-guided character control has emerged as a promising paradigm that bridges semantic expressiveness and physical realism, addressing limitations of purely data-driven text-to-motion synthesis that often produces artifacts like foot sliding or implausible contacts. Physics-simulated characters enforce physical plausibility (Juravsky et al., 2022; 2024; Yao et al., 2024; Truong et al., 2024; Tevet et al.; Wu et al., 2025), with PADL showing natural language as a direct control interface from simple instructions (Juravsky et al., 2022) to complex multi-skill tasks (Juravsky et al., 2024), MoConVQ leveraging pretrained motion codebooks and LLMs (Yao et al., 2024), PDP combining diffusion with physics-based imitation (Truong et al., 2024), and CLOSD introducing closed-loop plan-and-imitate architectures (Tevet et al.). Together, this line of work suggests a unifying framework that integrates linguistic flexibility with physical fidelity in a closed-loop system, motivating our approach.

**Real-world Humanoid Whole Body Control.** Real-world humanoid whole-body control has progressed rapidly with sim-to-real RLFu et al. (2025); Cheng et al. (2024); Ji et al. (2024), teleoperationHe et al. (2024; 2025), and large-scale retargeting(Yin et al., 2025; ?). Collectively, these

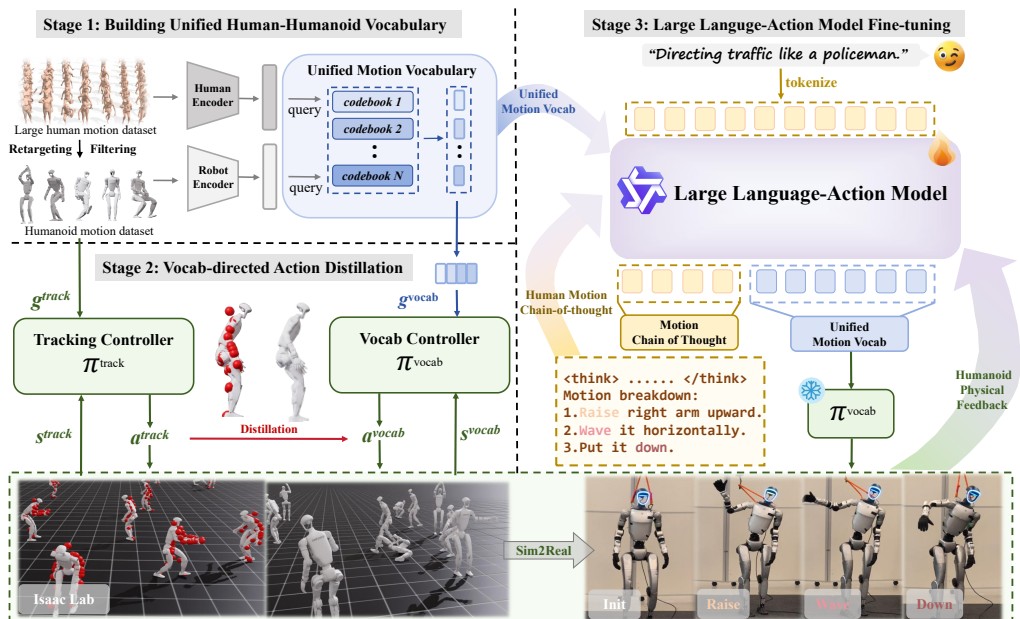

Figure 2: An overview of Humanoid-LLA. In stage one, we build a unified motion vocabulary leveraging a large-scale paired human and humanoid motion dataset. With a kinematic humanoid motion goal and its corresponding vocab retrieval, we distill a vocab-directed humanoid student controller from a teacher tracking controller. The first two stages enable stage three to acquire various humanoid feedback directly from physical simulation without decoding, making our LLA enhanced with high physical fidelity and language generalization.

advances provide strong controllers and data pipelines, yet most approaches decouple motion retargeting and control, leaving the semantic-to-physical generation gap open.

Language-conditioned humanoid control tackles this gap by directly linking natural language to whole-body policies. Large-scale efforts like UH-1 (Mao et al., 2024) and ALMI (Shi et al., 2025) advance text–motion corpora and hierarchical tracking but face challenges in real-world deployment. Shao et al. Shao et al. (2025) and RLPF (Yue et al., 2025) enhance policy learning with language mapping and physics feedback, yet remain limited by language generalization, conservative rewards, and reduced motion diversity. Overall, these studies highlight the need for unified frameworks that combine strong language generalization with the ability to generate diverse, expressive motions that are physically consistent.

## 3 METHOD

Our framework consists of three tightly connected components: **building unified human–humanoid motion vocabulary** (Sec. 3.1), **distilling vocabulary-directed policy** (Sec. 3.2), and **fine-tuning large language-action model** (Sec. 3.3). The first two components serve as essential prerequisites that make the integrated reasoning in the third component possible. Next, we introduce each component, highlighting its role within the overall framework.

### 3.1 UNIFIED HUMAN-HUMANOID VOCABULARY

**Humanoid Motion Canonicalization.** For human motion, prior work commonly adopts SMPL parameters to form a 263-dimensional representation (Loper et al., 2015; Guo et al., 2022), which serves as the learning target for generative models. To establish compatibility, we construct an analogous canonical representation for humanoid motion. Starting from the Unitree G1's (Robotics) raw control state $\mathbf{q} \in \mathbb{R}^{T \times 36}$ (including root translation, orientation, and joint DoF values), we apply a mapping $f : \mathbb{R}^{36} \to \mathbb{R}^{227}$ that augments kinematic details such as root velocities, 3D joint positions, and joint velocities. Each frame is thus represented as a structured 227-dimensional

vector, normalized to a root-centered coordinate system. This canonicalized form aligns with the human representation, enabling subsequent learning of a unified motion space.

**Implicit Partitioning Tokenization.** We aim to learn a unified tokenizer that maps human and retargeted humanoid motions into the same discrete vocabulary, ensuring that identical tokens carry consistent semantics across modalities. The tokenizer is expected to capture heterogeneous motion distributions while remaining compact for integration with language models. To this end, we adopt a VQ-VAE (Van Den Oord et al., 2017) with implicit partitioning (Ma et al., 2025), where each latent vector is split into sub-blocks and quantized by separate codebooks. Concatenating these assignments yields a large effective vocabulary without requiring a single oversized codebook. Beyond standard self-reconstruction within each modality (Zhao et al., 2025), we additionally enforce cross-modal reconstruction, such that a token obtained from either modality is decoded into the same motion primitive. This constraint ensures that identical tokens correspond to equivalent human and humanoid motions, thereby establishing a semantically unified motion representation.

**Cross-embodiment Optimization.** We optimize the dual-branch VQ-VAE by combining intra-modal and cross-modal reconstruction objectives. A sequence of human motion $\mathbf{m}^{\text{h}} \in \mathbb{R}^{T \times d_h}$ and humanoid motion $\mathbf{m}^{\text{r}} \in \mathbb{R}^{T \times d_r}$ are first encoded into latent features $\mathbf{z}^{\text{h}} = \mathcal{E}_{\text{human}}(\mathbf{m}^{\text{h}})$ and $\mathbf{z}^{\text{r}} = \mathcal{E}_{\text{robot}}(\mathbf{m}^{\text{r}})$, which are partitioned into sub-blocks and quantized by multiple codebooks to yield discrete tokens $\hat{\mathbf{z}}^{\text{h}}$ and $\hat{\mathbf{z}}^{\text{r}}$. These tokens are then decoded back to the motion space by modality-specific decoders $\mathcal{D}_{\text{human}}$ and $\mathcal{D}_{\text{robot}}$, producing both self-reconstructions ($\hat{\mathbf{m}}^{\text{h}}$, $\hat{\mathbf{m}}^{\text{r}}$) and cross-reconstructions ($\hat{\mathbf{m}}^{\text{r}\leftarrow\text{h}}$, $\hat{\mathbf{m}}^{\text{h}\leftarrow\text{r}}$). The additional cross-modal reconstruction enforces that the same token decodes into an equivalent motion across modalities, which is critical for achieving unified tokenization.

The training objective is defined as

$$\mathcal{L} = \mathcal{L}_{\text{intra}} + \alpha \mathcal{L}_{\text{commit}} + \beta \mathcal{L}_{\text{cross}}, \tag{1}$$

where $\mathcal{L}_{\text{intra}}$ is the intra-modal reconstruction loss for human and humanoid motions, $\mathcal{L}_{\text{cross}}$ penalizes discrepancies in cross-modal reconstruction (human-to-humanoid and humanoid-to-human), and $\mathcal{L}_{\text{commit}}$ is the commitment loss. Balancing coefficients $\alpha$ and $\beta$ control the trade-off between fidelity and codebook consistency. Architectural and training details are provided in Appendix C.1.

## 3.2 Vocabulary-directed Humanoid Action Distillation

With unified motion vocabulary in Sec.3.1, we next bridge the gap between kinematic motion primitives and physical control through a vocabulary-directed distillation process. Following the teacher–student paradigm used in recent whole-body controllers(He et al., 2025; Yin et al., 2025; Tessler et al., 2024), we train a privileged teacher policy to track continuous humanoid-retargeted motions with high fidelity and then distill its behavior into a vocabulary-directed student policy that relies on motion tokens. This stage shifts the control input from dense reference trajectories to the compact motion language of tokens, enabling the humanoid to execute token sequences output by the language model in Sec.3.3.

**Fully-constrained Teacher Controller.** We follow the goal-conditioned reinforcement learning framework to train a fully-constrained teacher tracking policy $\pi^{\text{track}}$ that tracks dense humanoid-retargeted reference states. At timestep $t$, the controller observes humanoid proprioception $\mathbf{s}_t$ and a goal state $\mathbf{g}_t^{\text{track}}$ comprising kinematic reference motion, and computes target joint positions $\mathbf{a}_t$ for the PD controller.

The teacher proprioception $\mathbf{s}_t$ consists of the current root linear velocity $\dot{\mathbf{p}}_t^{\text{root}} \in \mathbb{R}^3$, root angular velocity $\omega_t^{\text{root}} \in \mathbb{R}^3$, joint positions $\mathbf{q}_t \in \mathbb{R}^{n_j}$, joint velocities $\dot{\mathbf{q}_t} \in \mathbb{R}^{n_j}$ and the previous action history $\mathbf{a}_{t-1} \in \mathbb{R}^{n_j}$ with respect to the robot's local coordinate frame:

$$\mathbf{s}_t = \left[ \dot{\mathbf{p}}_t^{\text{root}}, \ \omega_t^{\text{root}}, \ \mathbf{q}_t, \ \dot{\mathbf{q}}_t, \ \mathbf{a}_{t-1} \right]. \tag{2}$$

And for tracking goal observation $\mathbf{g}_t^{\text{track}}$, we track relative body pose instead of absolute poses following previous tracking framework (Liao et al., 2025):

$$\mathbf{g}_t^{\text{track}} = \left[ \hat{\mathbf{q}}_{t+1}, \ \hat{\dot{\mathbf{q}}}_{t+1}, \ \hat{\mathbf{p}}_{t+1}^{\text{root}} - \mathbf{p}_t^{\text{root}}, \ \hat{\theta}_{t+1}^{\text{root}} \ominus \theta_t^{\text{root}} \right], \tag{3}$$

where $\ominus$ denotes the difference between two rotations. The policy action $\mathbf{a}_t$ is the normalized robot target joint positions, which are residual targets for nominal joint configuration.

For policy training, Proximal Policy Optimization (PPO) (Schulman et al., 2017) algorithm is used to maximize the accumulated reward $\mathbb{E}[\sum_{t=1}^{T} \gamma^{t-1} r_t]$. We design the reward $r_t$ as a weighted sum of task rewards, regularization and penalty. Details can be found in Appendix C.2.

**Vocabulary-directed Student Controller.** After fully-constrained teacher controller is trained, we distill $\pi^{\text{track}}$ into a vocabulary-directed student policy. Let the unified tokenizer (Sec. 3.1) provide a motion vocab window $\hat{\mathbf{z}}_{1:T}^{\text{vocab}}$, we aim to train a student policy $\pi^{\text{vocab}}$ that can generate full body actions satisfying these given motion vocabulary commands. To solve this ambiguity, we follow Tessler et al. (2024; 2025) and model $\pi^{\text{vocab}}$ as a Conditional Vatiational Autoencoder (CVAE) (Kingma & Welling, 2013) consisting of a vocabulary prior $\rho$, a residual encoder $\mathcal{E}$ and an action decoder $\mathcal{D}$. At timestep $t$, the motion vocab observation of the student controller is:

$$\mathbf{g}_t^{\text{vocab}} = \left[ \mathcal{M}(\mathbf{g}_t^{\text{track}}), \ \hat{\mathbf{z}}_t^{\text{vocab}} \right], \tag{4}$$

where $\mathcal{M}(\cdot)$ is a random masking function and $\hat{\mathbf{z}}_t^{\text{vocab}}$ is the current motion vocabulary in Sec. 3.1. The vocabulary prior is modeled as a Gaussian distribution over latents given the observed vocab constraints:

$$\rho(z_t | \mathbf{s}_t, \mathbf{g}_t^{\text{vocab}}) = \mathcal{N}\left( \mu^\rho(\mathbf{s}_t, \mathbf{g}_t^{\text{vocab}}), \ \sigma^\rho(\mathbf{s}_t, \mathbf{g}_t^{\text{vocab}}) \right). \tag{5}$$

The encoder $\mathcal{E}$ is modeled as a residual to the prior that outputs a latent distribution given the full-constraint teacher observation $\mathbf{g}_t^{\text{track}}$ (Yao et al., 2022):

$$\mathcal{E}(z_t | \mathbf{s_t}, \mathbf{g}_t^{\text{track}}) = \mathcal{N}\left( \mu^\rho(\mathbf{s}_t, \mathbf{g}_t^{\text{vocab}}) + \mu^{\mathcal{E}}(\mathbf{s}_t, \mathbf{g}_t^{\text{track}}), \ \sigma^{\mathcal{E}}(\mathbf{s}_t, \mathbf{g}_t^{\text{track}}) \right). \tag{6}$$

Based on the Dataset Aggregation (DAgger) algorithm (Ross et al., 2011), we train $\pi^{\text{vocab}}$ from $\pi^{\text{track}}$ with motion token labels within the same motion dataset. The training objective is to minimize the difference between reference action and student action as well as the KL divergence between encoder distribution $p_{\mathcal{E}}$ and prior distribution $q_\rho$:

$$\mathcal{L}_{\pi^{\text{vocab}}} = \|a_t^{\text{track}} - a_t^{\text{vocab}}\|_2^2 + \lambda_{\text{KL}}\left( p_{\mathcal{E}}(z_t | \mathbf{s_t}, \mathbf{g}_t^{\text{track}}) \ \| \ q_\rho(z_t | \mathbf{s}_t, \mathbf{g}_t^{\text{vocab}}) \right), \tag{7}$$

where $a_t^{\text{track}}$ is the reference action from $\pi^{\text{track}}$, $a_t^{\text{vocab}}$ is the student action sampled from $\mathcal{D}(a_t^{\text{vocab}} | \mathbf{s}_t, \mathbf{g}_t^{\text{vocab}})$ and $\lambda_{\text{KL}}$ is the hyperparameter for balancing reconstruction and regularization. Details can be found in appendix C.2.

### 3.3 LARGE LANGUAGE-ACTION MODEL

In this section we show how, building upon Sec. 3.1 and Sec. 3.2, our framework implements an end-to-end mapping from open-vocabulary and highly abstract language descriptions to physically executable robot actions without relying on tracking-based retargeting. Sec. 3.2 serves as the key intermediate: a low-level controller distilled to follow latent motion tokens, seamlessly linking latent motion token generation and physics-based action execution. The following parts in this section detail the training of our proposed LLA.

**Supervised Fine-tuning with Augmented Human Data.** We formulate motion token generation as an autoregressive, text-conditioned language modeling task, where a motion sequence is represented as a series of discrete tokens from the unified codebook $\mathcal{Z} = \{\langle cb_{i,j} \rangle\}$, with $i$ indexing the sub-codebook and $j$ the token entry. Given abundant paired human motion–text data, the input is the textual description $\mathbf{w}$, and the supervision target $\mathbf{y} = (y_1, \ldots, y_L)$ is constructed by concatenating a MLLM-annotated (Bai et al., 2025) motion chain-of-thought (Shao et al., 2024) with the ground-truth motion tokens from the pretrained tokenizer. The model is trained with the standard next-token prediction loss:

$$\mathcal{L}_{\text{SFT}} = -\mathbb{E}_{(\mathbf{w}, \mathbf{y}) \sim \mathcal{D}} \sum_{t=1}^{L} \log P_\phi(y_t \mid \mathbf{w}, y_{<t}), \tag{8}$$

where $\phi$ are the model parameters. This supervised stage establishes a preliminary alignment between language and motion, while enabling the model to respond by progressing from concise motion descriptions to richer analytical decomposition and ultimately to motion token generation.

**RL Fine-tuning with Humanoid Feedback.** Large models are commonly adapted to downstream tasks with reinforcement learning, resulting in policies that better match task-specific requirements. We adopt Group Relative Policy Optimization (GRPO) (Shao et al., 2024), a variant of PPO (Schulman et al., 2017) that avoids training a separate critic by sampling a group of candidate outputs $y^{(1:K)}$ for each input prompt $x$, assigning each a scalar reward, and normalizing rewards within the group to obtain relative advantages. This encourages the policy to prefer better-than-average candidates without requiring an explicit value function. The policy is optimized with a clipped surrogate objective regularized toward a reference model:

$$\mathcal{L}_{\text{GRPO}}(\phi) = -\,\mathbb{E}_x\,\mathbb{E}_{y^{(1:K)} \sim \pi_\phi}\left[\frac{1}{K}\sum_{k=1}^{K}\min\Big(r_k\,\tilde{A}_k,\ \text{clip}(r_k; 1-\epsilon, 1+\epsilon)\,\tilde{A}_k\Big)\right] + \beta_{\text{KL}}\,\mathcal{L}_{\text{KL}}, \quad (9)$$

where $x$ is the input prompt, $y^{(1:K)}$ are $K$ sampled candidate sequences, $r_k$ is the likelihood ratio between the current and reference policies, and $\tilde{A}_k$ is the group-normalized advantage. The KL term $\mathcal{L}_{\text{KL}}$ constrains the policy to stay close to a reference model. This formulation provides a stable and efficient way to fine-tune LLA with humanoid feedback, injecting physical priors into token generation.

Unlike prior work that emphasizes only kinematic fidelity (Ouyang et al., 2025; Yue et al., 2025), we stress the importance of dynamics-level consistency for real-world deployment. RLPF (Yue et al., 2025) employs a binary simulator-tracking reward, which ensures executability but often reduces motion diversity, as the policy tends to favor conservative behaviors that are easy to track. To address this, we design a reward scheme that combines high-level distributional objectives with low-level simulator-based tracking signals, achieving motions that are both physically robust and expressively varied.

**Physical Fidelity Reward Design.** The overall reward is a weighted sum of a binary format reward and a continuous physical fidelity reward. The format reward acts as a prerequisite: the model must first learn *how to answer* (i.e., producing valid structured outputs) before it can effectively learn *how to answer well* (i.e., generating physically and semantically aligned motions). Concretely, the format reward checks two requirements: (i) the response must follow a structured template beginning with `<think>...</think>` and followed by `<motion>...</motion>`; and (ii) within the motion segment, motion tokens must appear in cyclic sub-codebook order (`cb0→cb1→...→cb(N-1)` repeatedly). We define it as

$$r_{\text{format}} = \mathbb{I}\{\text{requirements satisfied}\}. \quad (10)$$

The physical fidelity reward is composed of a distributional term and a tracking term. The distributional reward encourages decoded motions to match the distribution of feasible trajectories and to align semantically with the paired text. Using contrastive encoders $\phi_{\text{m}}(\cdot)$ and $\phi_{\text{t}}(\cdot)$ (Guo et al., 2022) trained on physically plausible humanoid datasets, we define distributional reward as

$$r_{\text{dist}} = \exp\big(-\lambda_m\,\|\phi_{\text{m}}(\mathbf{m}_{\text{gen}}) - \phi_{\text{m}}(\mathbf{m}_{\text{ref}})\|_2\big) + \exp\big(-\lambda_t\,\|\phi_{\text{m}}(\mathbf{m}_{\text{gen}}) - \phi_{\text{t}}(\mathbf{w}_{\text{ref}})\|_2\big), \quad (11)$$

where the two terms measure motion fidelity and semantic fidelity, respectively, and $\lambda_m, \lambda_t > 0$ control sensitivity.

The tracking reward measures how well a generated token sequence can be executed in simulation by the distilled low-level controller (Sec. 3.2). We evaluate the simulated rollout with a position reward term $r_{\text{pos}}$ and an acceleration reward term $r_{\text{acc}}$:

$$r_{\text{track}} = r_{\text{pos}} + r_{\text{acc}} \quad (12)$$

Finally, the physical fidelity reward is calculated as $r_{\text{phys}} = r_{\text{dist}} + r_{\text{track}}$. More details are in appendix C.3.

## 4 EXPERIMENT

### 4.1 EXPERIMENT SETUP

**Dataset.** We conduct extensive experiments on the text-annotated subset of the AMASS dataset (Mahmood et al., 2019; Guo et al., 2022), consisting of 26,846 motion sequences, each

Table 1: Quantitative results on text-to-humanoid motion generation. We report R-Precision at top-3. ↑, ↓, and → indicate that higher is better, lower is better, and closer to the GT is better, respectively.

| Methods | FID↓ | R-Precision↑ | MM-Dist↓ | Div.→ |
|---|---|---|---|---|
| Ground Truth | 0.00 | 0.610 | 3.804 | 8.238 |
| MDM+Retarget (Tevet et al., 2023) | 11.759 | 0.262 | 6.599 | 6.419 |
| OmniH2O (He et al., 2025) | 17.159 | 0.222 | 8.021 | 5.868 |
| UH-1 (Mao et al., 2024) | 8.682 | 0.295 | 5.896 | 6.749 |
| LangWBC* (Shao et al., 2025) | 6.171 | 0.320 | 5.587 | 6.031 |
| Humanoid-LLA (Ours) | **2.626** | **0.447** | **4.911** | **7.122** |

Table 2: Physics-based quantitative results. ↑ and ↓ indicate that higher is better, lower is better, respectively.

| Methods | Succ.↑ | MPJPE↓ | $E_{vel}$ ↓ | $E_{acc}$ ↓ |
|---|---|---|---|---|
| OmniH2O (He et al., 2025) | 72.2% | 73.43 | 11.78 | 10.48 |
| UH-1 (Mao et al., 2024) | 68.8% | 121.51 | 16.59 | 14.80 |
| LangWBC* (Shao et al., 2025) | 76.0% | – | – | – |
| RLPF (Yue et al., 2025) | 80.0% | 140.00 | – | – |
| Humanoid-LLA (Ours) | **87.6%** | **56.43** | **8.92** | **7.74** |

paired with 3–4 textual descriptions. For every motion sequence, we employ `mink` (Zakka) to retarget human motions into corresponding humanoid motions, resulting in a paired human–humanoid dataset. The choice of this dataset is motivated by two factors. First, AMASS motions are captured using high-quality optical motion capture, ensuring low noise and enabling the model to better learn the latent alignment between motion and language. Second, text-annotated AMASS has been widely adopted in both human and humanoid motion generation, which ensures standardized and fair comparison across methods.

**Baselines.** To comprehensively demonstrate the advantages of our model in terms of both motion quality and physical executability for text-to-humanoid, we compare against several state-of-the-art baselines: 1) **MDM+Retarget** (Tevet et al., 2023) kinematically retargets MDM-generated motion to humanoid robots. 2) **OmniH2O** (Tevet et al., 2023; He et al., 2025) uses motion diffusion model to produce kinematic human motions followed by retargeting and an imitation policy to obtain physical humanoid motions. 3) **UH-1** (Mao et al., 2024) trains a decoder-only transformer to map text descriptions into humanoid motion with a retargeted humanoid motion-text dataset. 4) **LangWBC** (Shao et al., 2025) distills a VAE-based policy to simultaneously capture text semantics and sample actions. 5) **RLPF** (Yue et al., 2025) is a recent approach exploring physical feedback to constrain the kinematic LLM-based human motion generator, which is also followed by a post-process of motion retargeting and tracking. Implementation details of baselines are provided in appendix D.2. Besides text-to-humanoid, refer to appendix D for more experiments and ablation results for building unified motion vocabulary 3.1 and distilling vocab-directed controller 3.2.

**Evaluation Metrics.** Most prior work on text-to-humanoid motion generation (Mao et al., 2024; Shao et al., 2025; Shi et al., 2025; Yue et al., 2025) reports either low-level physics tracking metrics or human-motion generation metrics, leaving no unified protocol directly defined on humanoid robots. To fill this gap, we design an evaluation that combines physics-based tracking measures with distributional generation metrics computed in humanoid motion space. These two perspectives jointly capture executability, distributional fidelity, motion–language alignment, and diversity, thus discouraging models from producing only simple, easily executable motions at the expense of expressiveness. For the generation side, we report FID to measure distributional similarity against a physical humanoid motion set obtained by a goal-conditioned tracking policy (i.e., teacher controller in Sec. 3.2), MM-Dist and R-Precision to assess motion-language alignment, and Diversity (Div.) to evaluate variability. For the physics side, we measure success rate (Succ.), mean per-joint position error MPJPE (mm), velocity error $E_{vel}$ (mm/frame) and acceleration error $E_{acc}$ (mm/frame$^2$). More details are provided in appendix D.3.

Table 3: Quantitative results of ablation study.

| Methods | FID↓ | R-Precision↑ | MM-Dist↓ | Div.→ | Succ.↑ | MPJPE↓ | $E_{vel}$ ↓ | $E_{acc}$ ↓ |
|---|---|---|---|---|---|---|---|---|
| Humanoid-LLA w/o CoT | 10.423 | 0.270 | 6.222 | 6.405 | 64.90% | 90.43 | 14.11 | 11.23 |
| Humanoid-LLA w/o RLFT | 5.132 | 0.331 | 5.443 | 6.668 | 68.64% | 78.31 | 12.12 | 10.01 |
| Humanoid-LLA w/o $r_{dist}$ | 4.597 | 0.342 | 5.401 | 6.892 | 85.33% | 61.27 | 9.31 | 9.02 |
| Humanoid-LLA w/o $r_{track}$ | **2.578** | 0.439 | 5.013 | 7.007 | 76.72% | 66.42 | 10.89 | 9.77 |
| Humanoid-LLA (Ours) | 2.626 | **0.447** | **4.911** | **7.122** | **87.6%** | **56.43** | **8.92** | **8.74** |

## 4.2 TEXT-TO-HUMANOID EVALUATION

The results reveal distinct trade-offs among baselines. MDM (Tevet et al., 2023) generates motions in the human domain and transfers them to robots via kinematic retargeting, preserving expressiveness and diversity but lacking physical fidelity. OmniH2O (He et al., 2025) adds an imitation policy to obtain feasible trajectories, yet discrepancies between human and robot action spaces cause frequent tracking failures, and discarding these biases the motion distribution. UH-1 (Mao et al., 2024) trains on robot trajectories to decode from a robot-space latent manifold, improving fidelity and tracking scores while retaining generative capacity, but still falling short for real-world deployment. LangWBC (Shao et al., 2025) conditions on both language and control, achieving strong low-level executability but weaker motion–language alignment. RLPF (Yue et al., 2025) introduces physical feedback to constrain motions to the feasible set, but optimizing distributions in the human space yields suboptimal humanoid alignment.

In contrast, our method couples LLM-generated tokens with a vocabulary-directed controller and fine-tunes with humanoid feedback, preserving diversity and expressiveness while substantially boosting physical fidelity. This leads to consistent improvements across both evaluation axes, outperforming prior methods on generation metrics and tracking metrics. Implementation details see appendix C.

## 4.3 ABLATION STUDIES

We perform ablation studies to assess the contribution of each component of LLA in terms of generation quality and physical fidelity. (1) **Humanoid-LLA w/o CoT**: removes chain-of-thought augmentation and relies solely on raw motion descriptions when generating motion tokens. (2) **Humanoid-LLA w/o RLFT**: replaces the RL fine-tuned model with the SFT-only baseline. (3) **Humanoid-LLA w/o $r_{dist}$**: excludes the distributional reward while retaining the tracking-based term. (4) **Humanoid-LLA w/o $r_{track}$**: excludes the tracking reward while retaining the distributional term. The results highlight that each module plays a complementary role, and removing any of them leads to a clear degradation in performance.

## 5 CONCLUSION

In this work, We present Humanoid-LLA, a unified framework for language-conditioned humanoid control that bridges expressive language and huamanoid whole body execution. Our approach addresses the critical challenges of language generalization, physical fidelity and sim-to-real transfer in text-to-humanoid whole body motion generation. Specifically, Humanoid-LLA introduce a unified discrete codebook that aligns human and humanoid motion primitives, effectively bridging large language models and whole body controller. By augmenting large-scale human-motion datasets with vision language model generated annotations and fine-tuning with humanoid physics-based feedback in simulation, our model achieves enhanced language generalization and physical feasibility at execution. Extensive evaluations in physical environments demonstrate that our method outperforms prior works on both physical feasibility and motion quality, culminating in successful deployment on real humanoid hardware. Extending Humanoid-LLA to richer multimodal grounding, longer-horizon planning, and lightweight adaptation remains an important direction.

## 6 STATEMENTS

**Ethics statement**  We adhere to the ICLR Code of Ethics. This work uses publicly available research datasets and in-lab robot experiments conducted under standard safety protocols (e.g., emergency stop, clearance zones, and supervised operation). No human subjects research, personally identifiable information, or sensitive biometric data were collected. We followed all dataset licenses and terms of use, avoided revealing any private or proprietary content, and report results honestly and transparently.

**Reproducibility statement**  We will provide all materials needed to reproduce our results: training and evaluation code, configuration files with hyperparameters, environment specifications (OS, CUDA/driver, Python/package versions), random seeds, and scripts to download/preprocess datasets. We will release pretrained checkpoints, evaluation notebooks, and a README enabling end-to-end replication.

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

APPENDIX

## A  OVERVIEW

In this document, we provide expanded technical details, additional experiments, and extended discussions that complement and elaborate on the main paper. Specifically, Sec. C offers a detailed account of our implementation, covering the unified human–humanoid tokenization and codebook construction, the token-conditioned control policy with action distillation to torque-level actuation, and the Large Language–Action Model interface and training procedure; we also clarify settings required to reproduce baseline methods. Sec. D reports the robot system setup, metric definitions, and a comprehensive suite of experiments and ablations, and further illustrates generalization with additional textual results and motion visualizations; it also summarizes the accompanying video. Finally, Sec. E extends the discussion of limitations and failure cases and outlines directions for future work. Through this supplementary material, we aim to provide a more complete view of *Humanoid-LLA*, clarify practical nuances for replication, and furnish additional evidence of robustness and versatility.

## B  THE USE OF LARGE LANGUAGE MODELS (LLMS)

The Large Language Models were used only for English writing assistance such as grammar, wording, and minor stylistic edits to author-written text. The LLM did not contribute to research ideation, method design, experiments, analysis or citation selection. All technical content is authored and verified by the authors, who take full responsibility for the paper's contents. The LLM is not an author.

## C  SUPPLEMENTARY TECHNICAL DETAILS

### C.1  DETAILS OF UNIFIED HUMAN-HUMANOID TOKENIZATION.

**Humanoid Motion Canonicalization Details.**   We preprocess raw G1 humanoid trajectories with $T$ frames, each frame represented as

$$x_t = \left[ p_t \in \mathbb{R}^3, \ q_t \in \mathbb{R}^4, \ d_t \in \mathbb{R}^{29} \right], \qquad t = 0, \ldots, T-1. \tag{13}$$

---

**Algorithm 1** Humanoid Motion Canonicalization

---

1: **Input:** Raw trajectory $\{x_t\}_{t=0}^{T-1}$ with positions $p_t$, orientations $q_t$, DoFs $d_t$
2: **Output:** Canonicalized motion representation $\{f_t\}_{t=0}^{T-2}$
3: Downsample $\{x_t\}$ from $50\,\mathrm{Hz}$ to $20\,\mathrm{Hz}$    ▷ Resampling to align with human representation
4: **for** $t \leftarrow 0$ **to** $T-1$ **do**
5:   $r_t \leftarrow \mathrm{rotvec}(q_t)$
6:   $J_t \leftarrow \{j_t^k \in \mathbb{R}^3\}_{k=1}^{32}$ via FK
7: **end for**             ▷ Calculate forward kinematics for G1
8: $h_{\mathrm{floor}} \leftarrow \min_{t,k} j_{t,z}^k$
9: **for** $t \leftarrow 0$ **to** $T-1$ **do**
10:   $p_t \leftarrow p_t - [0,0,h_{\mathrm{floor}}]^\top$
11: **end for**             ▷ Floor alignment
12: **for** $t \leftarrow 0$ **to** $T-1$ **do**
13:   $p_t \leftarrow p_t - [p_{0,x}, p_{0,y}, 0]^\top$
14: **end for**
15: $a \leftarrow (j_0^{\mathrm{RHip}} - j_0^{\mathrm{LHip}}) + (j_0^{\mathrm{RShoulder}} - j_0^{\mathrm{LShoulder}})$
16: $\hat{a} \leftarrow a/\|a\|$
17: $\hat{f}_0 \leftarrow \dfrac{(0,0,1) \times \hat{a}}{\|(0,0,1) \times \hat{a}\|}$
18: $q_{\mathrm{align}} \leftarrow \mathrm{Quat}(\hat{f}_0 \mapsto +X)$
19: **for** $t \leftarrow 0$ **to** $T-1$ **do**
20:   $p_t \leftarrow \mathrm{Rot}(q_{\mathrm{align}})\, p_t$
21:   $q_t \leftarrow q_{\mathrm{align}} \otimes q_t$
22: **end for**             ▷ Root normalization
23: **for** $t \leftarrow 0$ **to** $T-2$ **do**
24:   $\omega_t \leftarrow r_{t+1} - r_t$             ▷ root angular velocity
25:   $v_t \leftarrow p_{t+1} - p_t$             ▷ root linear velocity
26:   $\Delta J_t \leftarrow J_{t+1} - J_t$            ▷ joint velocities
27:   $f_t \leftarrow [\omega_t, v_t, \mathrm{vec}(J_t), d_t, \mathrm{vec}(\Delta J_t)]$
28: **end for**            ▷ Canonicalized motion representation

---

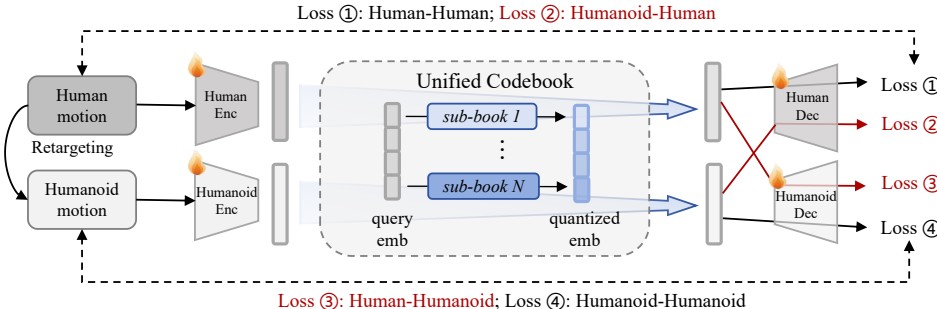

Figure A1: Diagram of detailed unified tokenizer architecture and training procedure.

We thus produce $(T-1)$ frames of 227-dimensional canonicalized humanoid motion representation per sequence (see Algorithm 1).

**Tokenizer Training Details.** To enforce embodiment-agnostic tokenization, we adopt a dual-branch VQ-VAE adapted from Zhang et al. (2023) where human and humanoid motions are encoded separately but quantized through shared codebooks. As shown in Fig. A1, given sequences $\mathbf{m}^{\mathrm{h}} \in \mathbb{R}^{T \times d_h}$ and $\mathbf{m}^{\mathrm{r}} \in \mathbb{R}^{T \times d_r}$, modality-specific encoders produce latents $\mathbf{z}^{\mathrm{h}}, \mathbf{z}^{\mathrm{r}}$, which are quantized into tokens $\hat{\mathbf{z}}^{\mathrm{h}}, \hat{\mathbf{z}}^{\mathrm{r}}$. Decoders then reconstruct both intra- and cross-modal motions. The corresponding objectives are

$$\mathcal{L}_{\mathrm{intra}} = \|\mathbf{m}^{\mathrm{h}} - \hat{\mathbf{m}}^{\mathrm{h}}\|_1 + \|\mathbf{m}^{\mathrm{r}} - \hat{\mathbf{m}}^{\mathrm{r}}\|_1, \quad \mathcal{L}_{\mathrm{cross}} = \|\mathbf{m}^{\mathrm{h}} - \hat{\mathbf{m}}^{\mathrm{h} \leftarrow \mathrm{r}}\|_1 + \|\mathbf{m}^{\mathrm{r}} - \hat{\mathbf{m}}^{\mathrm{r} \leftarrow \mathrm{h}}\|_1, \quad (14)$$

where cross-reconstruction ensures that shared tokens decode into semantically consistent motions across embodiments.

Each encoder–decoder is a temporal convolutional network with depth 3, dilation growth rate 3, and downsampling factor $2^2$. The latent space has 512 dimensions, evenly partitioned into 8 sub-chunks of 64 dimensions, each quantized by a codebook of size 64. Training is conducted on text-annotated AMASS with batch size 256. We use AdamW (lr=$2\times10^{-4}$, betas $(0.9, 0.99)$, weight decay $10^{-4}$) to optimize the tokenizer training. Both training and evaluation are run on a single NVIDIA RTX 4090 GPU.

### C.2 DETAILS OF VOCABULARY-DIRECTED ACTION DISTILLATION

**Details of Tracking Controller Reward Design.** As shown in Tab. A1, We train the fully con-strained teacher $\pi^{\text{track}}$ with PPO using a composite reward that combines normalized tracking terms (as exponential scores) with lightweight regularization and hard-limit penalties. Tracking targets are defined in the robot's local frame and computed as relative position and orientation error to reduce drift. Unless noted, all errors are normalized so that weights are comparable across terms.

Table A1: Reward table for the fully-constrained teacher controller

| Term | Weight | Term | Weight |
|---|---|---|---|
| **Task (tracking; $\exp(-\alpha\|\cdot\|_2)$ forms)** | | | |
| Root position | 0.5 | Root rotation | 0.5 |
| Body position | 1.0 | Body rotation | 1.0 |
| Body linear velocity | 1.0 | Body angular velocity | 0.5 |
| DoF position | 2.0 | DoF velocity | 0.2 |
| **Penalty (hard limits / self-contact)** | | | |
| Torque limits | $-1.0$ | DoF position limits | $-5.0$ |
| DoF velocity limits | $-5.0$ | Self-contact | $-0.1$ |
| **Regularization (L2 costs)** | | | |
| Lower-body action rate | $-0.4$ | Upper-body action rate | $-0.1$ |
| Torque | $-1\times10^{-4}$ | DoF acceleration | $-1\times10^{-5}$ |

**Domain Randomization.** To improve robustness under deployment, we adopt a broad range of randomization during policy training, including varying ground friction, joint damping, sensor la-tency, and external perturbations. With online adaptation of the teacher to randomized conditions, the distilled tokens encode transferable primitives rather than brittle overfits.

**Tracking Controller Training Details.** We train the tracking controller $\pi^{\text{track}}$ with on-policy PPO (Schwarke et al., 2025) using online data collection at a $50\,\text{Hz}$ control rate (physics $\mathrm{d}t = 0.005\,\text{s}$, action hold/decimation $= 4$) and $10\,\text{s}$ episodes in NVIDIA Isaac Lab (Mittal et al., 2023). Each iter-ation collects 24 policy steps per environment ($\approx 0.48\,\text{s}$ of experience) across up to 16,384 parallel environments on a flat plane, yielding up to $24 \times N_{\text{env}}$ transitions per update. To improve robustness, we apply domain randomization at startup (friction/restitution buckets, joint default pose pertur-bations, and anchor–body CoM shifts) and inject intermittent external pushes during rollouts by directly setting linear/angular velocities at random intervals between $1$–$3\,\text{s}$ (linear $\pm0.5\,\text{m/s}$ in $x/y$, $\pm0.2\,\text{m/s}$ in $z$; angular $\pm0.52\,\text{rad/s}$ roll/pitch, $\pm0.78\,\text{rad/s}$ yaw). Motion-conditioned commands are loaded from trajectories and sampled with an adaptive time-binning curriculum: the motion timeline is discretized into bins whose sampling probabilities are proportional to recent failure rates, smoothed with a short non-causal kernel (kernel size 3, $\lambda = 0.8$) and updated by an exponential moving average ($\alpha = 0.001$) with a small uniform mixture (ratio 0.1); when an episode terminates or a clip ends, time indices are resampled according to this distribution, and root/joint states are jittered and clipped to soft limits before continuing. PPO optimization uses clipping $\epsilon = 0.2$, learn-ing rate $10^{-3}$ with an adaptive schedule driven by a desired KL of 0.01, 5 learning epochs over 4 mini-batches per update, value loss coefficient 1.0 with value clipping enabled, entropy coefficient 0.005, discount $\gamma = 0.99$, GAE $\lambda = 0.95$, and max gradient norm 1.0; advantages use GAE and are

standardized. A low-frequency variant scales rollout length with the control period and exponentiates $\gamma$ and $\lambda$ to keep the effective per-second discount unchanged. All experiments are trained on 2 NVIDIA Geforce RTX 4090 48G GPUs with 30,000 iterations.

**Vocab Controller Training Details.** We implement the vocabulary-directed student as a conditional VAE whose components are: an encoder $\mathcal{E}$, a Transformer prior $\rho$, and an action decoder $\mathcal{D}$. At each step, the input is assembled from (i) masked tracking-goal poses (ii) 512-d motion vocabulary embedding, and (iii) a self-observation token. Each stream passes through normalized MLPs (clamp $= 5$) to produce 512-d tokens; visibility masks are mapped to the attention mask. The prior is a 4-layer, 4-head Transformer (feed-forward 1024, dropout 0.1) that parameterizes a Gaussian over the latent; the encoder provides a residual refinement MLP to this prior. The decoder is an MLP with layers 1024–1024–512 (ReLU, $\tanh$ head) that outputs normalized joint targets conditioned on the latent and current self-observation.

Training follows a teacher–student data-aggregation scheme: the student acts with masking enabled, a privileged tracking teacher supplies action labels, and we optimize an action reconstruction objective with a KL regularizer (annealed from $10^{-4}$ to $10^{-2}$ between epochs 3000 and 6000). We use 1024 parallel environments, 32-step rollouts, batch size 4096, 6 mini-epochs, Adam with learning rate $2 \times 10^{-5}$, and gradient clipping at 50.0. The target pose is visible with probability 0.1, and the vocab embedding is visible with probability 1.0 when present. At inference, latent noise is set to zero for deterministic control. All experiments are trained and inferenced on 2 NVIDIA Geforce RTX 4090 48G GPUs.

**Sim-to-Real Observation State Estimation.** Following prior works (Flayols et al., 2017; Liao et al., 2023; 2025) , we estimate root linear velocity $\dot{\mathbf{p}}^{\text{root}}$ by combining a momentum observer with an Extended Kalman Filter over base pose, velocity, and Inertial Measurement Unit biases. This filtering ensures that both teacher and student policies operate on physically plausible proprioception, closing the sim-to-real gap.

## C.3 DETAILS OF LARGE LANGUAGE-ACTION MODEL

**Human Data Augmentation.** Previous work (Ouyang et al., 2025) has highlighted that the sparsity and abstractness of text annotations in the AMASS dataset limit unified modeling of motion and language. Designing denser, decomposable, and more specific annotations can significantly enhance motion understanding. Motivated yet distinct from Ouyang et al. (2025), which employs LLMs to generate densified textual descriptions, we leverage the multimodal large model Qwen2.5-VL (Bai et al., 2025) to jointly process textual descriptions and rendered motion sequences. This enables us to obtain more accurate chain-of-thought (CoT) annotations, since a single high-level abstract description may correspond to multiple plausible motions, many of which do not align with the actual motion instance. These motion CoTs are then employed during the supervised fine-tuning stage to provide preliminary alignment between motions and language.

**Supervised Fine-tuning (SFT) Details.** We fine-tune Qwen2.5-3B-Instruct on our augmented human motion dataset. The original text annotations are used as part of prompts, while the motion Chain-of-Thought (CoT) together with the corresponding motion tokens serve as ground-truth responses. The model is trained autoregressively with cross-entropy loss. Training is conducted with batch size 32, learning rate $1 \times 10^{-4}$, weight decay 0.01, and the AdamW optimizer ($\beta = (0.9, 0.98)$) on 8 NVIDIA GPUs. We adopt a cosine scheduler with 100 warm-up steps, gradient clipping at 1.0, and mixed-precision training in bfloat16.

**Implementation of RL Fine-tuning.** We further fine-tune the model with reinforcement learning using the GRPO algorithm. Training is performed with batch size 64, consisting of 8 prompts per batch and 8 sampled responses per prompt, with gradient accumulation over micro-batches of 4. The maximum prompt and generation lengths are set to 512 and 1024, respectively. We use the memory-efficient AdamW optimizer with learning rate $1 \times 10^{-5}$, weight decay 0.01, and $(\beta_1, \beta_2) = (0.9, 0.999)$, along with gradient clipping at 1.0 and a cosine learning rate schedule decayed to $1 \times 10^{-6}$. The clipped objective adopts $\epsilon = 0.2$ and includes a KL regularization term with $\beta = 0.001$ against the SFT model as reference. The weighting coefficients in equation 11 are set as $\lambda_m = \lambda_t = 10$. We calculate $r_{\text{pos}} = \exp\left(-0.005\,\text{MPJPE}\right), r_{\text{acc}} = \exp\left(-0.05\,\text{E}_{\text{acc}}\right)$ in equation 12. The

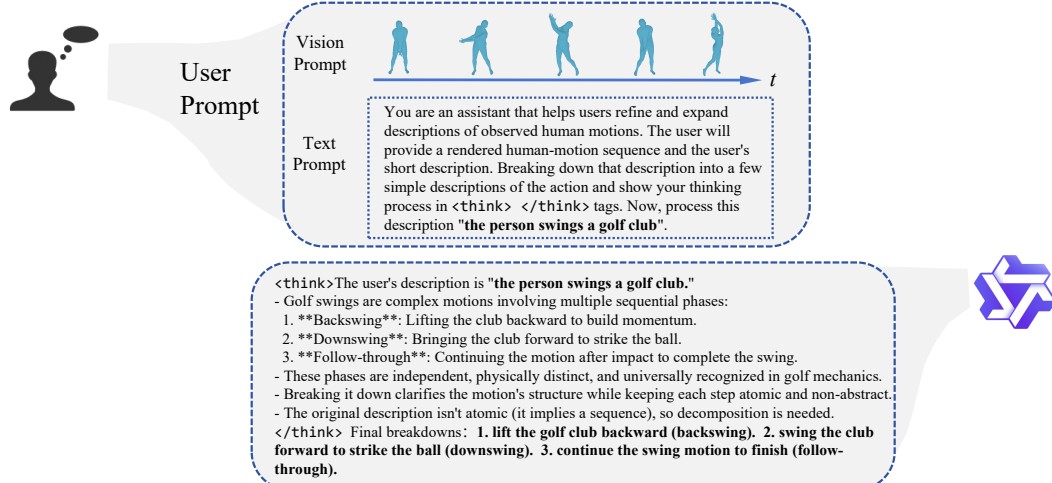

Figure A2: Visualization of Human Motion Chain of Thought augmentation based on Qwen2.5-VL.

contrastive motion encoder $\phi_m$ and text encoder $\phi_t$ in equation 11 are trained following Guo et al. (2022) on a tracking-based retargeted AMASS dataset, where only successfully tracked sequences are retained. The training is also conducted on 8 NVIDIA H20 GPUs.

# D    ADDITIONAL EXPERIMENTS AND RESULTS

## D.1    DETAILS OF ROBOT SYSTEM SETUP

Our real humanoid hardware is built on the Unitree G1 humanoid platform (Robotics), as shown in Fig. A3. The G1 stands 1320 mm tall, weighs about 35 kg with battery, and provides 6 DOF per leg and 7 DOF per arm, with a maximum arm payload of 3 kg. It is powered by a 9000 mAh detachable battery that supports around 2 hours of operation. The knee joint could achieve up to 139 N·m torque with a joint movement space of 0–165°, complemented by other flexible joints including the waist (Z: $\pm155°$, X: $\pm45°$, Y: $\pm30°$), the hip (P: $\pm154°$, R: $-30°$ to $+170°$, Y: $\pm158°$), and the wrist (P: $\pm92.5°$, Y: $\pm92.5°$), ensuring both stability and dexterity in whole-body control. We deploy our motion policies on the controller inference presented by (Liao et al., 2025). All deployment code is written in C++ and optimized for realtime execution, which achieved full-state estimation at 500 Hz using a low-level generalized momentum observer. The policy inference frequency is 100 Hz, enabling reliable real-time control of the robot during dynamic locomotion and manipulation tasks while ensuring smooth integration between state estimation and policy execution.

## D.2    IMPLEMENTATION OF BASELINES

For a fair comparison with the baselines introduced in Sec. 4, we unify the evaluation metrics and protocols across all methods. To measure generation metrics, we adopt the motion and text encoders described in C.3. The implementation details for each baseline are outlined as follows:

**MDM+Retarget**    We employ the MDM (Tevet et al., 2023) model pretrained on the HumanML3D (Guo et al., 2022) dataset to generate human motions. Since MDM outputs joint positions rather than SMPL parameters, we apply an IK-based optimization to regress SMPL parameters from joint positions, using a learning rate of $10^{-1}$ for 100 iterations per generated sequence. The resulting SMPL sequences are then retargeted to humanoid motions via an optimization-based method proposed in H2O (He et al., 2024). As this baseline does not involve physical simulation, we report only generation metrics.

**OmniH2O**    We implement OmniH2O (He et al., 2025) to track MDM+Retarget-generated humanoid motion sequences within a physics simulator. Following the evaluation protocol mentioned

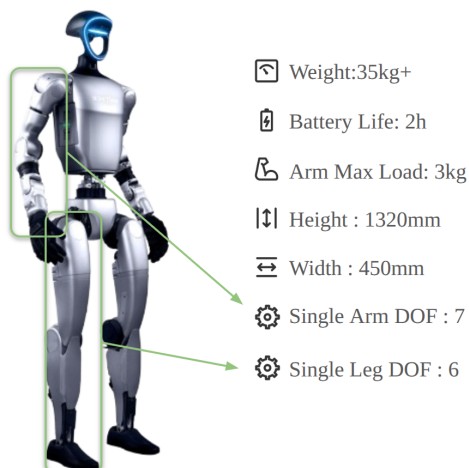

Figure A3: Details about Unitree G1 Robot.

in He et al. (2025), we report physics-based tracking metrics computed over all sequences, rather than only the successfully tracked ones. After tracking, all trajectories are collected to compute generation metrics.

**UH-1** We implement UH-1 (Mao et al., 2024) on our physically-retargeted humanoid motion dataset, which is collected by `mink` retargeting and our teacher controller tracking 3.2. We first train the humanoid motion generator based on T2M-GPT (Zhang et al., 2023), then leverage goal-conditioned RL to track these generated output. Generation metrics and physics metrics are reported following the same calculation paradigm as in in the implementation of OmniH2O.

**LangWBC** We reproduce LangWBC (Shao et al., 2025) by using a prior-free C-VAE student (no prior network) and a frozen CLIP text encoder (Radford et al., 2021) to jointly encode language and proprioception and decode normalized joint targets. For policy distillation we use the same teacher controller in Sec. 3.2.

**RLPF** For RLPF (Yue et al., 2025), we report the physics-based metrics of the RLPF-PHC variant, as its experimental setup and evaluation protocol are most comparable to other baselines. We do not include generation metrics, since RLPF evaluates them in the human motion domain, whereas our evaluation is defined directly on humanoid motions.

### D.3 DETAILS OF EVALUATION METRICS

We provide details of the metrics used for text-to-humanoid evaluation. Metrics are divided into two categories: generation-side metrics (Guo et al., 2022; Zhang et al., 2023), which measure semantic alignment and distributional fidelity, and physics-based tracking metrics (Luo et al., 2023), which assess physical executability in simulation.

Let $\mathcal{G} = \{(w_i^{(g)}, m_i^{(g)})\}_{i=1}^{N_g}$ denote the generated (text, motion) pairs, and $\mathcal{T} = \{(w_j^{(t)}, m_j^{(t)})\}_{j=1}^{N_t}$ denote the ground-truth test pairs. We denote the text encoder by $\phi_t(\cdot)$ and the motion encoder by $\phi_m(\cdot)$. Motion inputs are standardized before encoding using dataset mean and standard deviation:

$$\tilde{m} = \frac{m - \mu}{\sigma}, \qquad (15)$$

where $\mu, \sigma$ are the precomputed dataset mean and std.

**Generation-side metrics.**

- **Embeddings and pairwise distances.** For $N$ paired samples, compute text and motion embeddings:

$$t_i = \phi_t(w_i), \qquad m_i = \phi_m(\tilde{m}_i), \quad i = 1, \ldots, N, \tag{16}$$

and form the Euclidean distance matrix $D \in \mathbb{R}^{N \times N}$ as

$$D_{ij} = \sqrt{\max\left(-2\, t_i^\top m_j + \|t_i\|_2^2 + \|m_j\|_2^2,\ 0\right)}. \tag{17}$$

- **R-Precision@$k$.** Measures whether the ground-truth motion for each text query is among the $k$ nearest neighbors. Let $D_{i,(1)} \leq D_{i,(2)} \leq \cdots$ be the sorted distances in row $i$. Define

$$\mathbb{I}_{i,k} = \begin{cases} 1, & D_{i,i} \leq D_{i,(k)}, \\ 0, & \text{otherwise}, \end{cases} \tag{18}$$

where $D_{i,i}$ is the distance to its paired motion. Then

$$\text{R-Precision@}k = \frac{1}{N} \sum_{i=1}^{N} \mathbb{I}_{i,k}. \tag{19}$$

- **Matching score.** The average diagonal distance reflects text–motion alignment:

$$\text{MatchingScore} = \frac{1}{N} \sum_{i=1}^{N} D_{i,i}. \tag{20}$$

- **Diversity.** Measures intra-set variability of motion embeddings. Sample $T$ unordered pairs $\{(p_r, q_r)\}_{r=1}^{T}$:

$$\text{Diversity} = \frac{1}{T} \sum_{r=1}^{T} \|m_{p_r} - m_{q_r}\|_2. \tag{21}$$

- **Fréchet Inception Distance (FID).** Compares generated vs. ground-truth motion embedding distributions. For mean and covariance

$$\mu = \frac{1}{N} \sum_{i=1}^{N} m_i, \qquad \Sigma = \frac{1}{N-1} \sum_{i=1}^{N} (m_i - \mu)(m_i - \mu)^\top, \tag{22}$$

the FID is

$$\text{FID}(\mathcal{T}, \mathcal{G}) = \|\mu_t - \mu_g\|_2^2 + \text{Tr}(\Sigma_t + \Sigma_g - 2(\Sigma_t \Sigma_g)^{1/2}). \tag{23}$$

**Physics-based tracking metrics.**

- **Success rate (Succ).** Fraction of sequences tracked without falling or excessive deviation:

$$\text{Succ} = \frac{1}{N} \sum_{i=1}^{N} \mathbb{I}\left[\max_t \|J_t^{(i)} - \hat{J}_t^{(i)}\|_2 < 0.5 \text{ m}\right], \tag{24}$$

where $J_t^{(i)}$ and $\hat{J}_t^{(i)}$ denote reference and simulated joint positions.

- **Mean Per-Joint Position Error (MPJPE).** Average distance between predicted and reference joint positions:

$$\text{MPJPE} = \frac{1}{NTJ} \sum_{i=1}^{N} \sum_{t=1}^{T} \sum_{j=1}^{J} \|J_{t,j}^{(i)} - \hat{J}_{t,j}^{(i)}\|_2. \tag{25}$$

- **Velocity error ($E_{\text{vel}}$).** Discrepancy in per-joint velocities:

$$E_{\text{vel}} = \frac{1}{NTJ} \sum_{i=1}^{N} \sum_{t=1}^{T} \sum_{j=1}^{J} \|\dot{J}_{t,j}^{(i)} - \hat{\dot{J}}_{t,j}^{(i)}\|_2. \tag{26}$$

- **Acceleration error ($E_{\text{acc}}$).** Discrepancy in per-joint accelerations:

$$E_{\text{acc}} = \frac{1}{NTJ} \sum_{i=1}^{N} \sum_{t=1}^{T} \sum_{j=1}^{J} \|\ddot{J}_{t,j}^{(i)} - \hat{\ddot{J}}_{t,j}^{(i)}\|_2. \tag{27}$$

Table A2: Human motion reconstruction results. $N$ denotes the number of sub-codebooks, and $K$ the number of entries per sub-codebook. $\uparrow$, $\downarrow$, and $\rightarrow$ indicate that higher is better, lower is better, and closer to the dataset is better, respectively.

| Methods | FID$\downarrow$ | TOP-1$\uparrow$ | TOP-2$\uparrow$ | TOP-3$\uparrow$ | Diversity$\rightarrow$ | MM-Dist$\downarrow$ |
|---|---|---|---|---|---|---|
| T2M-GPT | $0.112^{\pm.001}$ | $0.500^{\pm.003}$ | $0.692^{\pm.002}$ | $0.789^{\pm.002}$ | $9.723^{\pm.066}$ | $3.056^{\pm.009}$ |
| Ours (N=4, K=64) | $0.077^{\pm.000}$ | $0.507^{\pm.003}$ | $0.699^{\pm.003}$ | $0.793^{\pm.003}$ | $9.645^{\pm.098}$ | $3.005^{\pm.008}$ |
| Ours (N=8, K=32) | $0.032^{\pm.000}$ | $0.509^{\pm.003}$ | $0.702^{\pm.002}$ | $0.796^{\pm.002}$ | $9.563^{\pm.063}$ | $2.986^{\pm.008}$ |
| Ours (N=8, K=128) | $0.018^{\pm.000}$ | $0.509^{\pm.002}$ | $0.702^{\pm.002}$ | $0.796^{\pm.002}$ | $9.579^{\pm.084}$ | $2.982^{\pm.009}$ |
| Ours (N=16, K=64) | $0.005^{\pm.000}$ | $0.510^{\pm.003}$ | $0.703^{\pm.003}$ | $0.797^{\pm.003}$ | $9.434^{\pm.069}$ | $2.968^{\pm.009}$ |
| Ours w/o $\mathcal{L}_{\text{cross}}$ | $\mathbf{0.041}^{\pm.000}$ | $\mathbf{0.508}^{\pm.003}$ | $\mathbf{0.700}^{\pm.002}$ | $\mathbf{0.794}^{\pm.002}$ | $\mathbf{9.488}^{\pm.072}$ | $\mathbf{2.987}^{\pm.009}$ |
| Ours (N=8, K=64) | $\mathbf{0.021}^{\pm.000}$ | $\mathbf{0.511}^{\pm.003}$ | $\mathbf{0.703}^{\pm.002}$ | $\mathbf{0.796}^{\pm.002}$ | $\mathbf{9.555}^{\pm.056}$ | $\mathbf{2.978}^{\pm.009}$ |

Table A3: Humanoid motion reconstruction evaluation results. $N$ denotes the number of sub-codebooks, and $K$ the number of entries per sub-codebook. $\uparrow$, $\downarrow$, and $\rightarrow$ indicate that higher is better, lower is better, and closer to the dataset is better, respectively.

| Methods | FID$\downarrow$ | TOP-1$\uparrow$ | TOP-2$\uparrow$ | TOP-3$\uparrow$ | Diversity$\rightarrow$ | MM-Dist$\downarrow$ |
|---|---|---|---|---|---|---|
| T2M-GPT | $0.183^{\pm.002}$ | $0.475^{\pm.003}$ | $0.661^{\pm.002}$ | $0.758^{\pm.002}$ | $10.804^{\pm.093}$ | $3.425^{\pm.008}$ |
| Ours (N=4, K=64) | $0.082^{\pm.001}$ | $0.484^{\pm.003}$ | $0.673^{\pm.002}$ | $0.771^{\pm.002}$ | $10.805^{\pm.096}$ | $3.338^{\pm.008}$ |
| Ours (N=8, K=32) | $0.037^{\pm.000}$ | $0.491^{\pm.003}$ | $0.679^{\pm.003}$ | $0.776^{\pm.002}$ | $10.577^{\pm.003}$ | $3.307^{\pm.008}$ |
| Ours (N=8, K=128) | $0.016^{\pm.000}$ | $0.492^{\pm.003}$ | $0.680^{\pm.003}$ | $0.777^{\pm.002}$ | $10.537^{\pm.094}$ | $3.291^{\pm.008}$ |
| Ours (N=16, K=64) | $0.006^{\pm.000}$ | $0.492^{\pm.003}$ | $0.681^{\pm.003}$ | $0.778^{\pm.003}$ | $10.653^{\pm.080}$ | $3.288^{\pm.008}$ |
| Ours w/o $\mathcal{L}_{\text{cross}}$ | $\mathbf{0.011}^{\pm.000}$ | $\mathbf{0.492}^{\pm.003}$ | $\mathbf{0.681}^{\pm.003}$ | $\mathbf{0.778}^{\pm.003}$ | $\mathbf{10.631}^{\pm.093}$ | $\mathbf{3.285}^{\pm.008}$ |
| Ours (N=8, K=64) | $\mathbf{0.023}^{\pm.000}$ | $\mathbf{0.490}^{\pm.003}$ | $\mathbf{0.678}^{\pm.003}$ | $\mathbf{0.776}^{\pm.002}$ | $\mathbf{10.671}^{\pm.089}$ | $\mathbf{3.301}^{\pm.008}$ |

## D.4 Additional Experiments and Ablations

**Experiments and Ablation on Unified Tokenizer.** To demonstrate the effectiveness of our implicit-partitioning tokenizer for fine-grained joint quantization of human and humanoid motion, we compare against T2M-GPT (Zhang et al., 2023), a representative baseline in motion quantization. We further ablate the number of sub-codebooks and the number of entries per sub-codebook, and we evaluate the effect of omitting the cross-reconstruction loss $\mathcal{L}_{\text{cross}}$. The evaluation of human motion reconstruction, humanoid motion reconstruction, human-to-humanoid motion reconstruction are all based on the implementation of Zhang et al. (2023). We show these experimental results in table A2, A3 and A4.

Results show that increasing the number of sub-codebooks and enlarging per-codebook capacity reduce quantization error. Compared with single-codebook quantization, implicit partitioning produces a more fine-grained discrete latent space under the same total token budget. As reported in Table A4, removing the cross-reconstruction term $\mathcal{L}_{\text{cross}}$ in Eq. equation 1 substantially degrades the human-to-humanoid reconstruction metric, demonstrating that the cross-modal objective is essential for assigning the same discrete token to semantically equivalent motion primitives across embodiments. Based on these ablations, we adopt 8 sub-codebooks with 64 entries each for our unified motion tokenizer; this configuration serves as the foundation for the motion–language joint modeling in Sec. 3.3.

**Ablations on Vocabulary-directed Action Distillation** Table A5 studies three key components of the vocabulary-directed student: a VAE latent, a Transformer prior, and random mask training. Replacing the **VAE** to MLP causes the largest drop in executability and accuracy: success falls from 95.2/86.1% (train/test) to 93.8/84.6%, while MPJPE degrades markedly ($39.86 \rightarrow 59.24$ train; $49.69 \rightarrow 68.57$ test), and $E_{\text{acc}}/E_{\text{vel}}$ increase (train: $6.88/6.13 \rightarrow 8.31/7.37$; test: $9.23/8.18 \rightarrow 11.73/10.84$). Dropping the **prior** also hurts but less severely: success 94.1/85.3% and MPJPE 48.72/59.91, indicating the prior supplies helpful dynamics regularization under token guidance. Eliminating the **mask** leads to the lowest test success (83.5%) and higher errors (MPJPE 61.83, $E_{\text{acc}} = 9.98$, $E_{\text{vel}} = 9.02$), suggesting that masking mitigates overfitting to dense teacher signals

Table A4: Human-to-Humanoid motion reconstruction evaluation results. $N$ denotes the number of sub-codebooks, and $K$ the number of entries per sub-codebook. $\uparrow$, $\downarrow$, and $\rightarrow$ indicate that higher is better, lower is better, and closer to the dataset is better, respectively.

| Methods | FID$\downarrow$ | TOP-1$\uparrow$ | TOP-2$\uparrow$ | TOP-3$\uparrow$ | Diversity$\rightarrow$ | MM-Dist$\downarrow$ |
|---|---|---|---|---|---|---|
| T2M-GPT | $0.381^{\pm.001}$ | $0.460^{\pm.002}$ | $0.642^{\pm.002}$ | $0.741^{\pm.002}$ | $10.689^{\pm.082}$ | $3.540^{\pm.007}$ |
| Ours (N=4, K=64) | $0.227^{\pm.002}$ | $0.468^{\pm.003}$ | $0.655^{\pm.003}$ | $0.754^{\pm.003}$ | $10.687^{\pm.083}$ | $3.445^{\pm.008}$ |
| Ours (N=8, K=32) | $0.182^{\pm.002}$ | $0.470^{\pm.003}$ | $0.657^{\pm.003}$ | $0.755^{\pm.002}$ | $10.777^{\pm.109}$ | $3.450^{\pm.007}$ |
| Ours (N=8, K=128) | $0.107^{\pm.001}$ | $0.476^{\pm.002}$ | $0.665^{\pm.002}$ | $0.764^{\pm.002}$ | $10.704^{\pm.099}$ | $3.387^{\pm.007}$ |
| Ours (N=16, K=64) | $0.084^{\pm.001}$ | $0.480^{\pm.003}$ | $0.668^{\pm.003}$ | $0.766^{\pm.002}$ | $10.701^{\pm.060}$ | $3.377^{\pm.007}$ |
| Ours w/o $\mathcal{L}_{cross}$ | $\mathbf{25.044^{\pm.028}}$ | $\mathbf{0.074^{\pm.001}}$ | $\mathbf{0.138^{\pm.002}}$ | $\mathbf{0.192^{\pm.002}}$ | $\mathbf{6.230^{\pm.048}}$ | $\mathbf{8.192^{\pm.009}}$ |
| Ours (N=8, K=64) | $\mathbf{0.153^{\pm.002}}$ | $\mathbf{0.477^{\pm.002}}$ | $\mathbf{0.665^{\pm.002}}$ | $\mathbf{0.762^{\pm.002}}$ | $\mathbf{10.736^{\pm.102}}$ | $\mathbf{3.396^{\pm.006}}$ |

Table A5: Ablations on Vocabulary-directed Action Distillation.

| Methods | HumanML3D-Train | | | | HumanML3D-Test | | | |
|---|---|---|---|---|---|---|---|---|
| | Succ $\uparrow$ | MPJPE $\downarrow$ | $E_{acc}$ $\downarrow$ | $E_{vel}$ $\downarrow$ | Succ $\uparrow$ | MPJPE $\downarrow$ | $E_{acc}$ $\downarrow$ | $E_{vel}$ $\downarrow$ |
| **Ours Tracking Controller** | **95.2%** | **39.86** | **6.88** | **6.13** | **86.1%** | **49.69** | **9.23** | **8.18** |
| Ours w/o VAE | 93.8% | 59.24 | 8.31 | 7.37 | 84.6% | 68.57 | 11.73 | 10.84 |
| Ours w/o prior | 94.1% | 48.72 | 8.19 | 7.28 | 85.3% | 59.91 | 9.56 | 8.71 |
| Ours w/o mask | 92.9% | 51.32 | 8.62 | 7.58 | 83.5% | 61.83 | 9.98 | 9.02 |
| **Humanoid-LLA (Ours)** | **95.0%** | **46.84** | **8.04** | **6.86** | **87.6%** | **56.43** | **8.92** | **7.74** |

and encourages reliance on discrete tokens. Overall, the vocabulary-directed student approaches the teacher's success while accepting slightly higher pose errors. After integrating the student into the full RL pipeline, **Humanoid-LLA** surpasses the teacher in test success (**87.6**% vs. $86.1$%) and further reduces $E_{acc}/E_{vel}$ to $8.92/7.74$.

## D.5 ADDITIONAL TEXT-TO-HUMANOID VISUALIZATION

We include extra visualization results in the simulation and the real world in this material. More visualization can be found in the supplementary video.

## E EXTENDED LIMITATION AND DISCUSSIONS

In this work, we present the first end-to-end Large Language–Action Model for physical-fidelity, open- vocabulary humanoid control, mapping expressive natural language directly to executable humanoid actions. Through comparative experiments, we identify considerable areas where our model can be further improved, as outlined below:

**Longer-Horizon Memory and Planning.** Our LLA reasons over token sequences within a modest temporal window, which limits plan consistency across complex tasks. A natural extension is to couple Humanoid-LLA with a hierarchical planner that maintains a persistent memory (e.g., key–value token cache or episode summaries) and proposes subgoals that the vocabulary-directed controller can realize, improving stability and global coherence over minutes-long activities.

**Richer Multimodal Grounding.** We condition primarily on text (and optionally visual renders), while real deployments benefit from audio cues (speech prosody), gaze/pose of humans, and tactile events. Extending the tokenizer with cross-modal slots (speech/vision/touch tokens) could align linguistic intent with environmental context, enabling disambiguation.

**Personalization and Style Control.** Different users may prefer distinct motion styles or safety margins. Conditioning tokens on user embeddings (or few-shot style exemplars) can produce personalized motions while preserving safety. A style–safety Pareto controller could expose interpretable dials (conservativeness, speed, energy) without retraining.

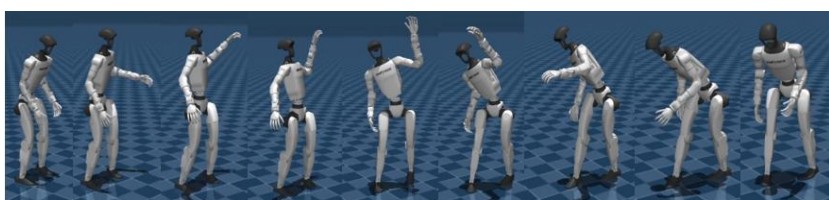

"A person swings his left arm over head as if he is spiking a volleyball."

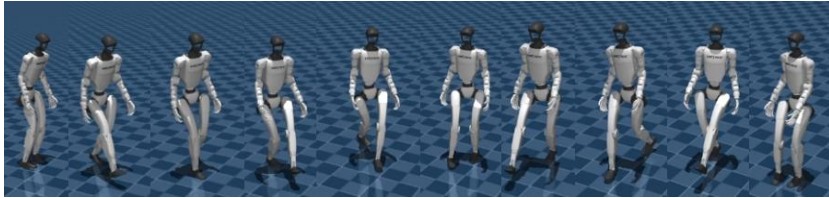

"Walk along a curving path, adjusting direction smoothly."

Figure A4: Visualization results in Mujoco.

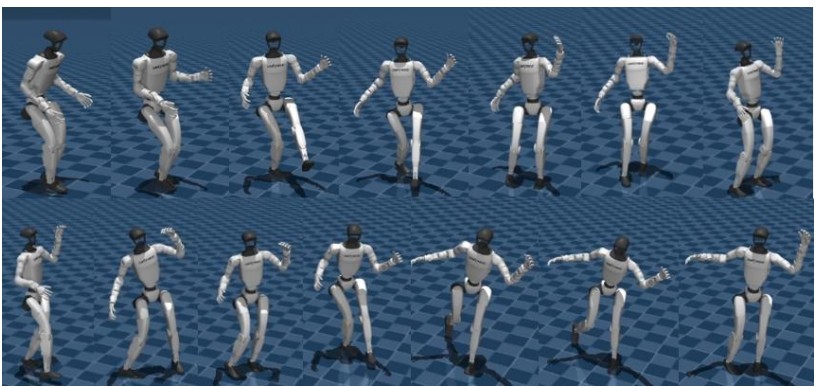

"Step in wide arcs while circling slowly, arms gently outstretched for balance,
creating a smooth flowing spiral path."

Figure A5: Visualization results in Mujoco.

**Scaling Data and Benchmarks.** Our unified tokenizer aligns human and humanoid motions; scaling paired data with richer captions and hard negative text–motion pairs should improve semantic precision. We also advocate benchmarks that jointly score distributional quality and physics on robots, preventing degenerate solutions that optimize only one axis.

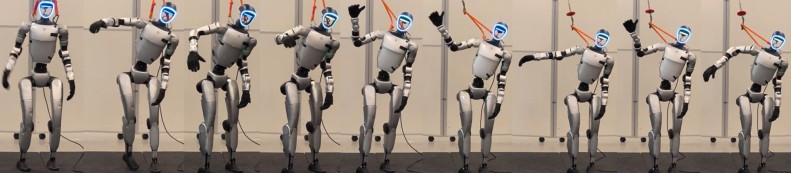

"Directing traffic like a policeman."

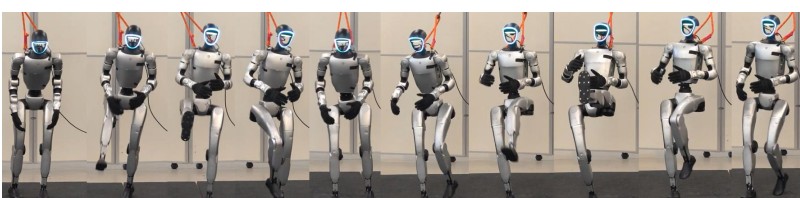

"Goose step forward like a soldier."

Figure A6: Visualization results in the real world.

