# OpenReview forum: "Humanoid-LLA: Open-Vocabulary Humanoid Whole-Body Control with Large Language Action Model"
_ICLR.cc/2026/Conference — ICLR 2026 Conference Withdrawn Submission_

### Official Review · Reviewer_cZg7 · 2025-10-31

**Soundness:** 3
**Presentation:** 3
**Contribution:** 3
**Rating:** 6
**Confidence:** 3

**Summary:**

This paper introduces Humanoid-LLA, a unified framework for open-vocabulary humanoid whole-body control. The goal is to map natural language instructions to physically executable whole-body motions for humanoid robots, going beyond the manipulation-focused Vision-Language-Action (VLA) models. Experiments on the AMASS dataset and a real Unitree G1 humanoid demonstrate strong open-vocabulary generalization and high physical fidelity. Quantitative results show consistent improvements over OmniH2O, UH-1, LangWBC, and RLPF across both language–motion alignment and physics-based metrics.

**Strengths:**

1.	The paper provides a coherent three-stage pipeline that bridges semantic language representations and physically grounded control. The introduction of a unified motion vocabulary is novel and crucial for aligning human and robot action spaces in a discrete, LLM-friendly format.
2.	The paper evaluates both generation quality (FID, R-Precision, Diversity) and physical executability (MPJPE, velocity/acceleration errors). Ablation studies (removing CoT, RLFT, rdist, rtrack) demonstrate the contribution of each component clearly. Comparisons span across representative paradigms (diffusion, VAE, RL), showing generality.
3.	The method is deployed on a physical humanoid (Unitree G1) and claims real-world feasibility.

**Weaknesses:**

1.	During upper-body motion execution, the lower body does not maintain stable standing posture — is this because the whole-body controller has not been sufficiently trained yet? The movement still appears somewhat shaky.
2.	The quality of motions in the AMASS dataset varies considerably. What criteria were used to select and post-process the subset utilized for training? This inconsistency might also increase the difficulty of training the subsequent tracking controller.
3.	It might be helpful to provide qualitative comparisons with other existing methods to further demonstrate the effectiveness of the proposed approach.

**Questions:**

See Weakness

---

### Official Review · Reviewer_9KXX · 2025-10-31

**Soundness:** 3
**Presentation:** 3
**Contribution:** 3
**Rating:** 4
**Confidence:** 4

**Summary:**

This paper presents Humanoid-LLA, a large language–action model that enables open-vocabulary, language-conditioned whole-body control for humanoid robots. The approach builds a unified human–humanoid motion vocabulary, a vocabulary-directed controller distilled from a physics-based teacher, and a reinforcement learning fine-tuning stage with dynamics-aware rewards to ensure both semantic alignment and physical feasibility. Trained on large-scale text–motion datasets with chain-of-thought augmentation, Humanoid-LLA achieves superior motion naturalness, stability, and real-world robot execution compared to prior language-conditioned controllers

**Strengths:**

- This paper is well written and well motivated. It tackles a challenging problem of mapping language instructions to real humanoid motion.
- The proposed unified human–humanoid vocabulary is promising and novel.
- The experiments are extensive and include evaluations in both simulation and the real world.

**Weaknesses:**

- The related work section overlooks important literature on text-to-motion generation, such as [1] and [2], which demonstrate complex simulated humanoid motions [1,2] and real-world deployment on a quadrupedal robot [2]. The differences and advantages of this paper compared to these existing works should be discussed more thoroughly.
- The real-world motions demonstrated are relatively simple and limited in number, which raises an important question: Is such a complex LLA framework truly necessary? Is a large pre-trained model essential as the backbone network? What if the model only selected from a small set of predefined basic motions, or if a language-conditioned BC policy were trained instead? While I agree that open-vocabulary capability is important for future development, more challenging tasks would be needed to convincingly demonstrate its necessity.
- There is no ablation study comparing performance with and without discrete motion tokens. While such experiments may be difficult due to the current pipeline design, at least some discussion should be provided. For instance, could a language model directly encode open-vocabulary instructions into embeddings to condition a BC policy?
- The main text is not fully self-contained and should be expanded, especially regarding the LLA component. In Appendix C.3, the authors mention using Qwen2.5-VL to jointly process textual descriptions and rendered motion sequences. It remains unclear how these rendered sequences are obtained during evaluation and how they influence final performance. What would happen if only an LLM without visual input were used?
- In the reproducibility statement, the authors promise to provide all materials and code, but none are included in the supplementary files.

[1] AnySkill: Learning Open-Vocabulary Physical Skill for Interactive Agents. CVPR 2024

[2] LAGOON: Language-Guided Motion Control. ICRA 2024

**Questions:**

See the weaknesses part. Overall, I think this paper is interesting and timely. I would like to raise my score if the author could address these concerns convincingly.

---

### Official Review · Reviewer_LEGh · 2025-11-01

**Soundness:** 3
**Presentation:** 3
**Contribution:** 2
**Rating:** 4
**Confidence:** 4

**Summary:**

This paper introduces Humanoid-LLA, a framework for open-vocabulary, language-conditioned humanoid whole-body control. The key insight is to represent motion through a unified human–humanoid motion vocabulary that aligns human motion data with humanoid actuation constraints in a shared discrete token space. The authors evaluate on text-to-humanoid motion benchmarks derived from AMASS and show that Humanoid-LLA substantially outperforms prior methods (e.g., UH-1, LangWBC, RLPF) in both motion–language alignment (FID, R-Precision) and physical fidelity (success rate, MPJPE). The system also demonstrates deployment on real humanoid hardware (Unitree G1).

**Strengths:**

- **Novel unified motion representation:** The idea of a *shared human–humanoid motion vocabulary* is conceptually strong and technically well-motivated. It directly addresses the data scarcity and embodiment mismatch problem that limits language-conditioned humanoid control.
- **End-to-end language-to-control pipeline:** The integration of language modeling, motion tokenization, and torque-level control into a single architecture (via distillation) represents a meaningful step beyond prior “text-to-human-motion + retargeting” pipelines.
- **Well-structured multi-stage training:** The three-stage pipeline (vocabulary learning → policy distillation → RL fine-tuning) is logically coherent and experimentally validated.
- **Comprehensive evaluation:** The experiments combine generative metrics (FID, R-Precision) and physics-based metrics (success rate, MPJPE, velocity/acceleration errors), enabling a balanced assessment of realism and physical feasibility.
- **Ablation and analysis:** The ablations (removing CoT, RLFT, rdist, or rtrack) clearly demonstrate the contribution of each module.

**Weaknesses:**

- **Dependence on retargeted motion data:** Despite the stated goal of “bypassing retargeting,” the unified vocabulary and teacher policy are still built upon retargeted human-to-humanoid motion pairs. The framework still inherits biases and errors from the retargeting stage, meaning the open-vocabulary generalization is bounded by the retargeted motion space.
- **Limited real-world performance:** The real-world performance is limited and not so impressive. And also no real-world or simulation qualitative comparison with at least one baseline, which is perhaps more intuitive to see whether the method is useful.
- **Lack of clarity on motion token interpretability:** It remains unclear whether the discrete tokens correspond to meaningful or composable motor primitives, or if they are merely latent representations. Deeper analysis on this will make the paper more meaningful.
- **Ambiguity in open-vocabulary evaluation:** While the term “open-vocabulary” is emphasized, it is not explicitly shown how the system handles genuinely novel language instructions unseen in training, or compositional generalization beyond motion rephrasing.

**Questions:**

What's the real-world execution latency of this method compared to other baselines?

---

### Official Review · Reviewer_HGNB · 2025-11-01

**Soundness:** 4
**Presentation:** 4
**Contribution:** 2
**Rating:** 2
**Confidence:** 4

**Summary:**

The paper Humanoid-LLA introduces a unified Large Language-Action Model (LLA) that enables humanoid robots to execute open-vocabulary natural-language commands as physically feasible, whole-body motions

The approach integrates three key components:

1. a unified human-humanoid motion vocabulary learned via cross-modal VQ-VAE tokenization to align motion primitives across embodiments;
2. a vocabulary-directed controller distilled from a privileged physics-based teacher policy to map discrete tokens to executable control;
3. and a language-to-motion generator trained with SFT on text–motion datasets and further refined via RL with physics-based rewards.

Experiments on AMASS-based text-motion data and the Unitree G1 platform show that Humanoid-LLA achieves SoTA open-vocabulary humanoid control, improving motion fidelity and real-robot success rate over previous methods.

**Strengths:**

1. The paper introduces a dual-branch VQ-VAE with cross-modal reconstruction loss to jointly quantize human and humanoid trajectories into a shared discrete codebook. This design eliminates kinematic misalignment seen in prior retargeting pipelines and provides a reusable motion token space suitable for LLM reasoning.

2. A two-stage teacher–student process distills a privileged PPO tracking policy into a token-conditioned CVAE controller, ensuring physical feasibility while maintaining semantic control. This hierarchical abstraction is conceptually strong and empirically validated through improved success and lower MPJPE in Table 2.

3. RL Fine-Tuning with Physical Fidelity Rewards. Though GRPO is widely used for LLM post-training, this work adapt it to humanoid text-to-motion generation and proved effectiveness by using a designed composite rewards.

4. Comprehensive Evaluation and Ablations. The experiments compare against strong baselines such as LangWBC, UH-1, and RLPF in terms of FID and MPJPE The ablation table clearly quantifies the contribution of each component, validate methodological soundness.

5. Unlike many purely kinematic works, Humanoid-LLA is validated on real humanoid hardware (Unitree G1), confirming transferability from simulation and robustness under physical constraints.

**Weaknesses:**

Although the simulation metrics are strong, the real-world demonstrations suggest that physical stability remains a challenge. In the accompanying video, the humanoid displays noticeable oscillations during walking and running, and even brief balance losses (e.g., around 1:59, the robot falters while walking; around 2:10, it struggles to stand steadily while moving its arms). These behaviors indicate that while the proposed vocabulary-directed controller achieves promising physical feasibility in simulation, its real-world execution still lacks consistent balance and robustness. This gap suggests that further refinement of the controller or sim-to-real adaptation is needed to fully realize the paper’s vision of open-vocabulary humanoid control.

**Questions:**

1. Could the authors evaluate Humanoid-LLA’s real-world robustness under external perturbations, for instance, by introducing controlled pushes or uneven ground tests to quantify how well the vocabulary-directed controller maintains balance compared to LangWBC or other baselines?

2. The current hardware video contains partial and cropped clips where motion segments are interrupted. Could the authors provide a continuous, unedited sequence showing the robot completing full-body motions from start to finish to better evidence execution stability?

3. LangWBC presented a clear “Wave → Run → Wave” task demonstrating seamless transitions between expressive and dynamic behaviors. Could the authors reproduce a similar evaluation protocol to highlight whether Humanoid-LLA achieves comparable motion continuity and balance across distinct instruction types?

---

### Note · Authors · 2025-11-12

I have read and agree with the venue's withdrawal policy on behalf of myself and my co-authors.